# A porcine model of Fanconi anemia

Brandon Hergert[1], Kristin M. Whitworth[2], Devorah C. Goldman[1], Lisa Moreau[3], Kelsey McQueen[3], Kalindi Parmar[3], Alan D'Andrea[3], Melissa S. Samuel[2], Kevin D. Wells[2], Randall S. Prather[2], Craig Dorell[1], Markus Grompe[1], William H. Fleming[1]*

1 Department of Pediatrics, Papé Family Pediatric Research Institute, OHSU, Portland, Oregon,
2 Division of Animal Sciences, University of Missouri, National Swine Resource and Research Center, Columbia, Missouri, United State of America, 3 Dana Farber Cancer Institute, Harvard University, Boston, Massachusetts, United State of America

* flemingw@ohsu.edu

## Abstract

Although small animal models of Fanconi anemia (FA) are useful, they do not faithfully replicate many of the clinical features seen in FA patients. We reasoned that a porcine model of FA with its similar physiology and a relatively long lifespan would produce a phenotype more similar to human FA. Targeting *FANCA* in domestic swine resulted in skeletal abnormalities and extreme sensitivity to interstrand DNA cross-linking agents. In addition, FANCA disruption followed by mitomycin C treatment resulted in a > 10-fold increase in chromosomal radials, a finding that is considered diagnostic for human FA. Bone marrow derived, hematopoietic progenitor cells from a FANCA null pig showed a 75% reduction in colony forming activity compared to wild type. Evaluation of steady state hematopoiesis in the peripheral blood revealed the gradual development of red cell macrocytosis and a reduction in circulating neutrophils. Targeting of *FANCD2* failed to produce any biallelic animals demonstrating the loss of FANCD2 function is embryonic lethal in pigs. These results indicate that a porcine model of FANCA holds promise for the development of strategies to prevent the development of bone marrow failure and malignancies in patients with FA.

## Introduction

Fanconi anemia (FA) is a complex, autosomal-recessive disorder that disrupts a critical DNA repair protein pathway thereby preventing the repair of interstrand-crosslinks (ICLs) [1,2]. Patients with null or hypomorphic FA mutations can present with a variety of congenital and developmental abnormalities [3,4]. Throughout their life patients accumulate DNA damage particularly in tissues with a high proliferation rate. This leads to anemia, bone marrow failure and leukemia as well as squamous cell carcinomas [5,6]. Treatment of these disorders is particularly challenging in FA patients and animal models hold promise for the preclinical testing of novel therapeutic approaches [6].

**Data availability statement:** All relevant data are within the paper and its Supporting Information files.

**Funding:** U420D011140 National Heart Lung & Blood Inst. (NIH) RSP No # Fanconi Anemia Research fund MG and WHF Co-PIs.

**Competing interests:** The authors have declared that no competing interests exist.

To date 23 FA genes have been identified, 21 of which have a respective murine model [7,8]. These FA mice typically have the anticipated defects in DNA repair and show extreme sensitivity to DNA damage by alkylating agents as demonstrated by chromosomal breakage assays [9]. However, although hematopoietic stem cells (HSC) in fetal and adult FA mice show some subtle defects in self-renewal [10] FA mice do not spontaneously develop anemia, bone marrow failure, acute leukemia or squamous cell cancers [7]. This may in part be due to the relatively short lifespan of these mouse models. Despite being very useful for studying DNA repair pathways, FA mutant mice do not faithfully recapitulate the key clinical phenotypes seen in FA patients [11,12]. This significantly limits their utility as a platform to study therapeutic strategies aimed at preventing the most common complications of FA and other diseases.

Pigs are being increasingly used as large animal models because of their similar physiology and size in comparison to humans [13,14]. In addition, they have a relatively long lifespan that permits the accumulation of mutations in highly proliferative tissues including the bone marrow and epithelial tissues [15]. Here we report the creation of the first porcine FA model, using CRISPRCas9 technology to target the *FANCA* and *FANCD2* genes in domestic pig oocytes followed by in vitro fertilization. We subsequently bred *FANCA* edited pigs to create a biallelic F1 generation that shows evidence of the classic DNA repair phenotype, skeletal abnormalities and disrupted hematopoiesis.

## Materials and methods ethics statement

The use of animals during this study was in accordance with the approved protocol and standard operating procedures established by the Animal Care and Use Committee of the University of Missouri which strictly adheres to the Animal Welfare Act to assure limited discomfort and pain. The National Swine Resource and Research Center is AAALAC accredited to ensure excellent oversight of animal protocols and facilities.

Embryo transfer surgeries were performed with both preemptive and postoperative analgesics to alleviate pain. The analgesic used in this study was flunixin meglumine (Banamine) at a dose of 2.2 mg/kg. In addition to analgesics, pigs are closely monitored every 15 minutes post-surgery until sternal. Once sternal, pigs are monitored twice daily until the surgical case is closed. Isoflurane (2–5%) was the anesthesia used in this study. If additional pain management was required, flunixin meglumine (Banamine) and oral phenylbutazone were used. Pigs are euthanized with a commercially available euthanasia solution (euthasol) at a dose 1 ml/4.5 kg.

## Oocyte injection, in vitro fertilization, and embryo transfer

Ovaries from pre-pubertal gilts were obtained from an abattoir (Smithfield, Milan, MO). Cumulus oocyte complexes were aspirated from follicles and matured in vitro as described previously [16]. Mature oocytes were selected and the guide RNA + Cas9 RNA mix were co-injected into the oocyte cytoplasm by using a FemtoJet microinjector (Eppendorf; Hamburg, Germany). After injection, oocytes were

in vitro fertilized [17,18] and cultured until day 4 to day 6 post fertilization. Between 42 and 48 hours post fertilization morula and/or blastocysts were removed from the culture medium and were surgically transferred into the oviduct of the recipient gilt. The use of recombinant DNA was approved by the University of Missouri Institutional Biosafety Committee.

## Guide design and in vitro transcription of single guide RNAs

Two guides were designed in exon 4 of pig *FANCA* gene. Guide 1 (5'-GCAGGATCAAGCCTCGCGGT-3') and guide 3 (5'-GGATCTGTGCTTCGGACGCC-3'). Guides were screened for off target targets in NCBI blast. For screening in porcine fibroblasts, the guide

strands were cloned into pSpCas9(BB)-2A-GFP (PX458) [19], a gift from Feng Zhang (Addgene plasmid # 48138; http://n2t.net/addgene:48138; RRID:Addgene_48138). For in vivo targeting,

gBlock(Integrated DNA Technologies, Coralville, IA) containing each DNA guide sequence with an upstream T7 promoter sequence was designed to express the *FANCA* specific guide RNA. The gBlock was then used as template DNA for in vitro transcription as described previously [18,20]. Briefly, each gBlock was diluted to a final concentration of 0.1 ng/µl and PCR amplified with gBlock forward primer (5'-ACTGGCACCTATGCGGGACGAC-3') and a gBlock reverse primer (5'AAAAGCAC CGACTCGGTGCCAC-3') with Q5 (New England Biolabs, Ipswich, MA). The PCR conditions were an initial denaturation of 98°C for 1 min followed by 35 cycles of 98°C (10s), 68°C (30s) and 72°C (30s). Each amplicon was then purified by using a PCR purification kit (Qiagen, Valencia, CA) and used as a template for in vitro transcription with the MEGAshortscript T7 kit (ThermoFisher, Waltham, MA). The resulting RNA was then purified by using the MEGAclear Transcription Clean-Up Kit (ThermoFisher, Waltham, MA). *FANCA* RNA guides 1 and 3 were mixed with Cas9 RNA (TriLink Biotechnologies) at a final concentration of 20 ng/µl Cas9 RNA and 10 ng/µl of guide RNA. RNA was stored at -80C until immediately prior to zygote injection.

## Pig fibroblast cultures

Fibroblast primary cell cultures were derived from piglet tail skin shortly after birth. Tails were washed in 70% EtOH followed by removal of the skin which was then finely cut and digested for four hours at 39°C in DMEM medium containing 10% FBS, 1x anti-anti, 25 Kunitz DNase/mL and 200 Units/mL collagenase IV. The digestion was filtered through a 70 µm strainer and the remaining skin was further digested in 0.25% trypsin with EDTA for 30 minutes with intermittent inversion. This fraction was filtered again and combined with fraction one. Cells were then pelleted by centrifugation at 1500 rpm and resuspended in 10 mL supplemented DMEM and plated in a T75 flask at 39°C for 24 hours before replacing the medium. To validate FANCA guide strands in vitro, cells were nucleofected with PX458 plasmids expressing guide strands, Cas9 and GFP. Single GFP+ cells were sorted into individual wells of a 96 well plate and clones were established, expanded and cryopreserved. To generate a "corrected" mutant pig fibroblast culture, FANCA-/- pig fibroblasts were transduced with lentivirus encoding wildtype human FANCA (Rocket Pharma) with an MOI of 10:1.

## Polymerase chain reaction for mutations

Mutation burden was determined by PCR amplification of *FANCA* genomic at the CRISPR/Cas9 target site using forward primer (5'-TCGTGCAGAGCTGATAAAGTTC-3') and reverse primer (5'AGCTGTGGTAACCGAGGGT-3') flanking the target site to amplify a~900-bp region. PCR products were either purified and directly Sanger sequenced or TOPO cloned into the pCR4 vector (Invitrogen, Waltham, MA) and transformed into competent One Shot TOP10 *E. coli* (Invitrogen, Waltham, MA). Individual colonies were sampled for vector isolation and sequenced by Sanger sequencing to confirm genotypes. All gene editing experiments were approved by the Institutional Biosafety Committee.

## Blood Collection and Complete Blood Counts (CBCs)

CBCs were performed by the staff at the University of Missouri Veterinary diagnostic Laboratory under the supervision of Dr. Cheryl Rojas and Dr. Angela Royal. Blood was collected from the jugular vein into EDTA coated tubes periodically every few months. CBCs performed by the University of Missouri Veterinary Medical Diagnostic Laboratory using the Siemens Advia 2120i Hematology system.

## Fibroblast Colony Forming Assay

Fibroblasts originating from a biallelic *FANCA* knockout pig were tested for their sensitivity to DNA cross linking reagents by assessing their ability to form colonies in vitro [21] in the presence of low doses of diepoxybutane (DEB) or mitomycin C (MMC) when exposed to the alkylating agent. Primary pig fibroblasts were plated at 300 cells per 35 mm well while human WT (GM639) and human FA (GM6914) controls were plated at 150 cells per 35 mm well. Each well contained 2 ml of supplemented DMEM media as described previously. DEB was then serially diluted to 1 mg/ml, 0.1 mg/ml and 0.025 mg/ml followed by a 1000x dilution into the plated fibroblast media in duplicate for a concentration gradient starting at 0 µg/ml (PBS), 0.025 µg/ml and 0.1 µg/ml, then fixed using 0.5 ml of 70% EtOH for an additional 30 minutes. After fixation, the colonies were stained with 1 ml of 3% methylene blue overnight. Colonies with >15 cells were counted as "true colonies" and included in the data sets for statistical analysis.

## Chromosome Breakage Assay

Whole blood was harvested from animals at the NRRSC in sodium heparin coated 4 ml tubes. To expand lymphocytes, 0.7 ml of blood was cultured in 9.3 ml of RPMI 1640 containing 15% FBS 1% penicillin/streptomycin, 1% L-glutamine and 100 ul/10 ml PHA-M (Life Technologies, Carlsbad, CA) for 1 day and then treated for two days with 20 ng/ml mitomycin c (MMC).Colcemid (0.1 µg/ml) (Life Technologies, Carlsbad, CA) was added for 2 hours and then cells were harvested and centrifuged at 1200 RPM for 10 minutes. The supernatant was removed and 13 ml of warm (~37°C) 0.075 M KCl solution were added. The cells were then gently resuspended with a Pasteur pipette and then incubated for 20 minutes at 37°C. Next, 1 ml of fresh fixative (3parts methanol: 1-part acetic acid) was added followed by gentle inversion and centrifugation at 1200 RPM for 10 minutes. The supernatant was removed and 5–6 ml of fresh fixative was added and the sample was centrifuged for 10 minutes at 1200 RPM. Two additional washes in fixative were performed prior to making slides containing chromosomal spreads. Analysis of 50 metaphases per sample was performed.

## Hematopoietic colony forming assay

Hematopoietic colony forming assays were performed by seeding thawed pig bone marrow in Methocult H4034 (Stem Cell Technologies) at $10^5$ cells per ml in triplicate 35 mm dishes and scoring colonies after 13 days in a 37°C, 5% $CO_2$ incubator.

## Results

### Disruption of FANCA exon4 in pig fibroblasts produces an FA phenotype

Before creating *FANCA* exon 4 null pigs, we performed a proof-of concept experiment to evaluate the functional significance of FANCA exon 4 deletion in pig cells. Primary fibroblasts isolated from the tail of normal pigs were nucleofected with plasmids expressing Cas9, sgRNA targeting FANCA ex4 and a GFP marker [19]. Single GFP+ cells were FACS sorted into 96-well plates and expanded (**Fig 1A**). Clonal fibroblasts were evaluated for *FANCA* exon4 mutations by Sanger sequencing and Synthego ICE analysis [22]. Two independent biallelic clones, mutant A, FANCA exon 4 (+250/+220) and mutant B, FANCA exon 4 (+1/+2) were tested for their sensitivity to the DNA cross linking agent diepoxybutane (DEB), in fibroblast colony-forming assay forming assays. Both mutant clones showed significant hypersensitivity to DEB, in contrast to the wild type (WT) controls (**Fig 1B**). As these fibroblast clones were extensively

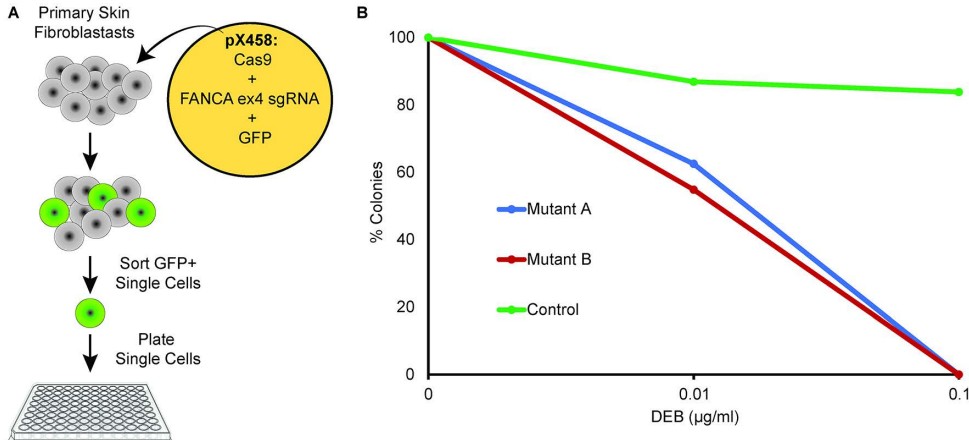

**Fig 1.** *In vitro* **targeting of FANCA exon 4 in primary pig fibroblasts. (A)** Fetal pig fibroblasts were nucleofected with CRISPR-Cas9 targeting FANCA exon 4 using sgRNA and a GFP marker. After 2 days of culture, GFP+ cells were cell-sorted into a 96-well plate expanded for 1 week and then geno-typed for FANCA exon 4 mutations. Three clones were selected for phenotypic analysis and tested for their sensitivity to DEB in colony forming assay; Control (WT/+1), Mutant A(+1/+2) and Mutant B(+250/+220). Both mutant clones showed clear hypersensitivity to DEB consistent with loss of FANCA function.

expanded primary *FANCA* -/- cells, an insufficient number of metaphases could be harvested to perform chromosomal breakage studies. Based on this finding of hypersensitivity to an alkylating agent in fibroblasts, we elected to proceed with targeting exon 4 in fertilized pig zygotes followed by embryo transfer (**Fig 2**).

## Creation of FANCA exon 4 edited pigs

FANCA exon 4 in pig oocytes was targeted using the CRISPR/cas9 system followed by in vitro fertilization and implanta-tion of targeted morula/blastocysts into surrogate females (Fig 2). Of the first 10 piglets born, 6 showed successful editing of exon 4 of the FANCA gene and 5 of the 6 had monoallelic edits (S1 Table). A single piglet had a biallelic edit, but one of the edited alleles contained a 3 bp in-frame deletion. Consequently 2 of the FANCA edited founder gilts, 115−1(WT,-3) and 115−4(WT,-2) were bred with a FANCA edited boar, 115−8(WT,+7) to create an F1 generation. The edits were aligned to Refseq (XM_005653229.3) and included a 2 bp GC deletion in exon 4 at location $\Delta$ 369 (−2), a 3 bp TGC deletion in exon 4 at location 368 (−3) and a 7 bp GCACCCA insertion at location $\Delta$ 363 (+7). Litter 67, derived from the breeding pair 115−4 X 115−8, produced 8 piglets, 5 of which were successfully targeted including the biallelic piglet 67−2 (+7,-2) (S1 Table).

## Evaluation of the phenotype of biallelic FANCA exon 4 targeted piglet

Piglet 67−2 harbored frame edits (+7,-2) in exon 4 that would be anticipated to functionally disrupt the *FANCA* gene, because both cause frame-shifts. Interestingly, this animal was born with one additional dewclaw on the radial aspect of each front leg (**Fig 3A**). The extra dewclaws are variable in size and radiographs show they have expected structure (**Fig 3B**) compared to the wild-type structure (**Fig 3C**). Importantly there was no impairment of this animal's gait or mobility. This condition is similar to midline preaxial polydactyly seen in some human FA patients [23]. There were no other physi-cal abnormalities detected at birth. At adulthood, tongue epithelium in the mutant animals showed delayed differentiation (**Fig 4A**) and increased proliferation (**Fig 4B**) relative to a control pig (**Fig 4 C&D**), including regions of mild dysplasia.

## Sensitivity to DNA crosslinking

To determine if the *FANCA* edited pig, 67−2 has the characteristic hypersensitivity to DNA crosslinking reagents, we exposed primary skin fibroblasts, isolated from the tail at birth, to DEB in a colony forming assay (**Fig 5A**). SV40

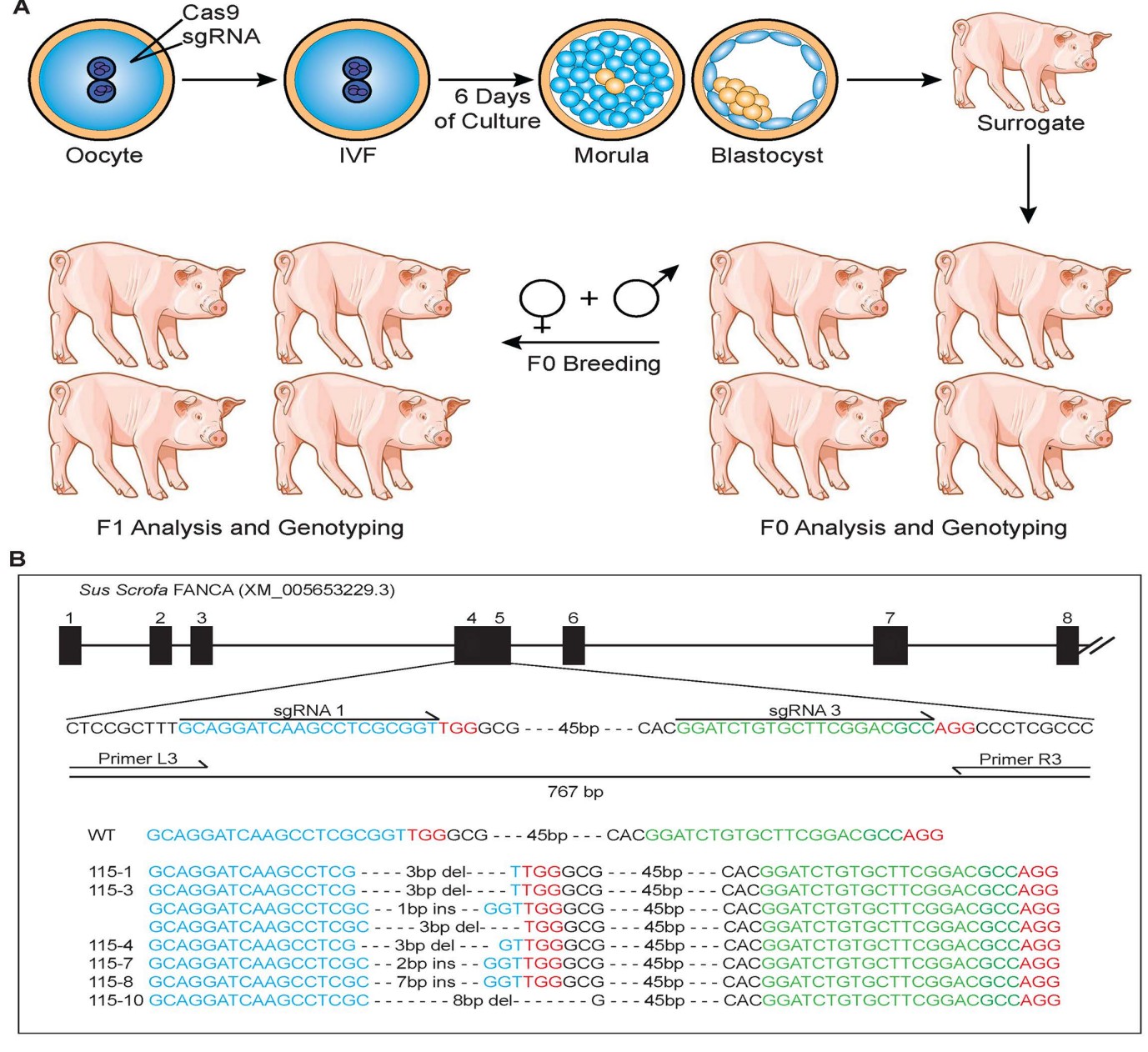

**Fig 2. Creation of FANCA KO pigs using CRISPR-Cas9 targeting. (A)** Pig oocytes were targeted for FANCA by microinjection of sgRNA and mRNA encoding Cas9 followed by in vitro fertilization. Zygotes were cultured to the blastocyst/morula stage then transferred to a surrogate mother. Initial progeny underwent genotypic and phenotypic analysis. As mosaicism often occurs, heterozygous animals were bred creating an F1 generation with biallelic out-of-frame FANCA pigs. **(B)** Schematic of Exon 4 targeting construct and examples of the resulting indel events.

transformed WT and FA human fibroblasts were used as controls. Similar to the human FA cell line GM6914, fibroblasts from the 67−2 pig showed a modest reduction in colony forming activity at a dose of 0.025 µg/ml DEB and an almost complete loss of colony forming activity at dose of 0.1 µg/ml of DEB (**Fig 5B**). Combined results from 3 independent experiments utilizing primary fibroblast cultures from 67−2 yielded similar results (**Fig 5C**).

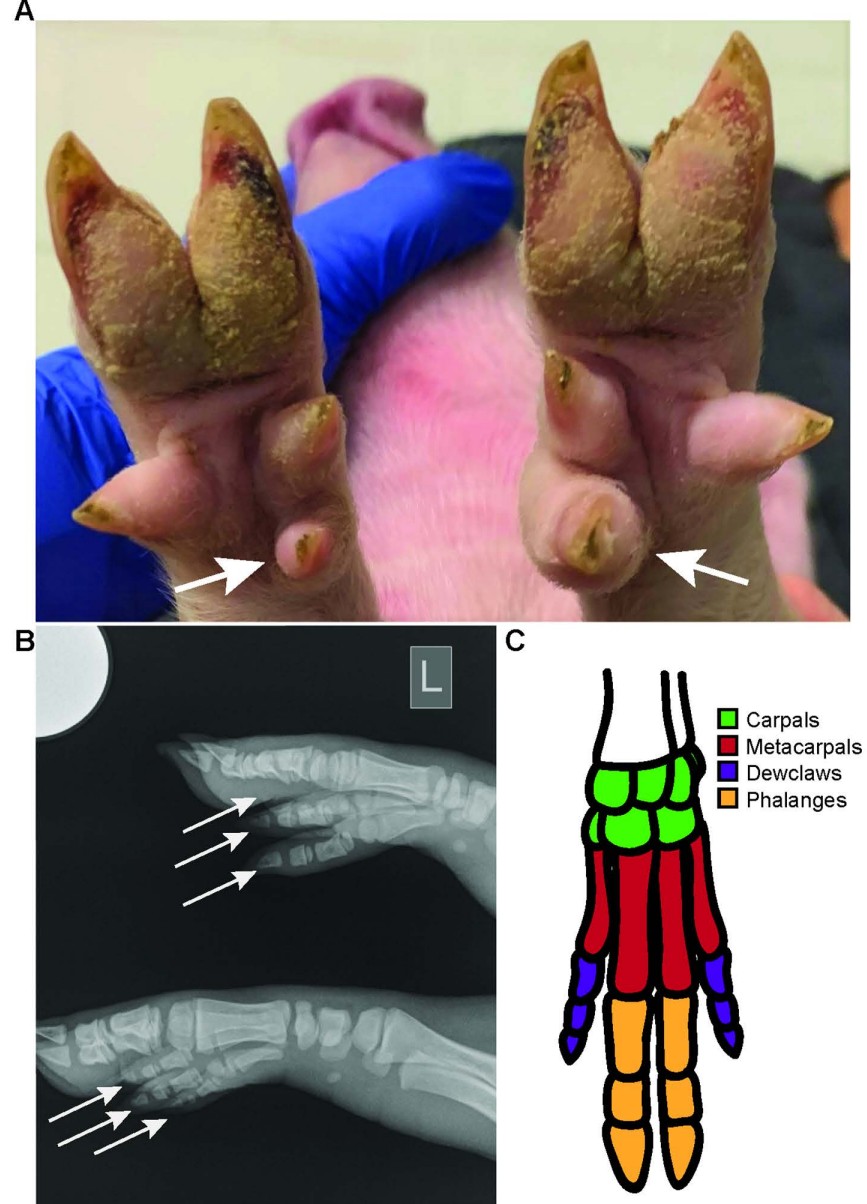

**Fig 3. Polydactyl dew claws in a FANCA exon 4 targeted piglet. A)** Image of front limbs showing extra dewclaws of variable size on the medial aspect of both front limbs of F1 generation piglet 67−2. **B)** X-rays of 67−2 showing an extra dew claws on (white arrows) both front limbs. **C)** Anatomy of normal porcine foot.

## Chromosome breakage assay

The chromosome breakage assay is the gold standard for human FA diagnosis [21,24] and similar results are seen in animal models of FA including mice and zebrafish [25,26]. Peripheral blood mononuclear cells (PBMC) from WT pigs and piglet 67−2 were cultured in the presence of the T-cell mitogen PHA and the DNA cross linking agent mitomycin C (MMC). Subsequently metaphase cells were harvested for karyotypic analysis [27]. Following exposure to MMC, metaphases from 67−2 displayed a > 10-fold increase in the number of chromosomal aberrations per cell compared to wild type controls

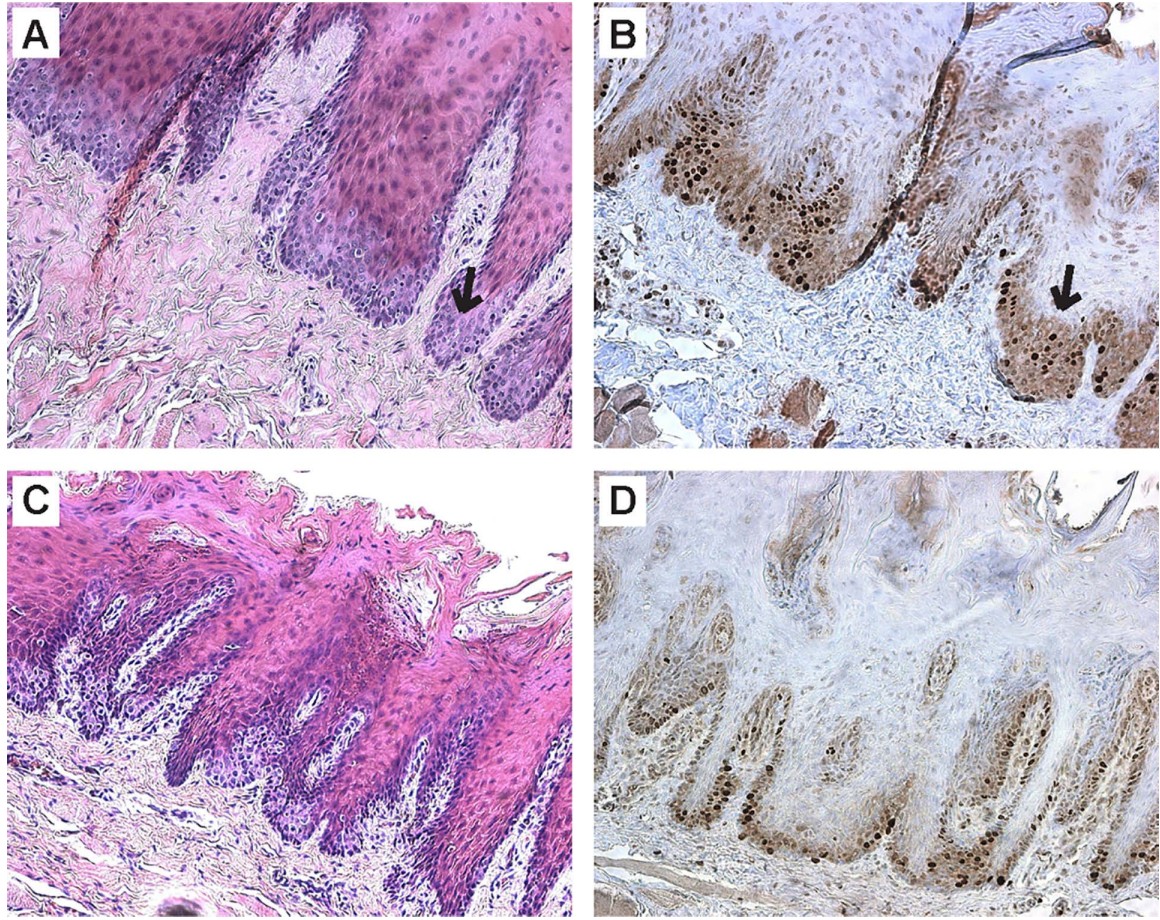

**Fig 4. FANCA-/- pig tongue histology is abnormal.** Mutant pig (A,B) and wild type (C,D) tongue sections were stained with H&E or labeled with Ki-67 to visualize proliferation. Arrows indicate areas of delayed differentiation. Total magnification 100x.

(Fig 6A) and more than 33% of the cells from 67−2 showed the formation of chromosomal radials (Fig 6B). Examples of chromosomal aberrations found in cells from piglet 67−2 are shown (Fig 6C&6D). This high frequency of chromosomal aberrations and radials is considered diagnostic of the DNA repair defect seen in humans with FA. Transduction of these mutant fibroblasts with a lentivirus encoding human *FANCA* dramatically normalized MMC sensitivity in a colony forming assay (Fig 6E). This finding confirms that the loss of FANCA activity and not an off-target effect was responsible for the DNA damage phenotype. Importantly, these results show that our porcine model is suitable for testing potential gene therapy approaches using the human FANCA gene to correct the FA phenotype.

## Hematologic profile

In the neonatal period, complete blood counts (CBC) from animal 67−2 were indistinguishable from its wild type and monoallelic littermates. All CBC parameters remained essentially within their normal ranges until 28 weeks of age when a clear increase in mean corpuscular hemoglobin concentration (MCH) was noted. By week 38 an increased red cell volume (MCV) was also observed. These parameters remained elevated through 63 weeks of age (Fig 7A). Also during this time frame, the absolute neutrophil count decreased by more than 50% below the baseline value and fell below the normal range for adult pigs. Red cell macrocytosis is commonly seen as an early manifestation of anemia in FA patients

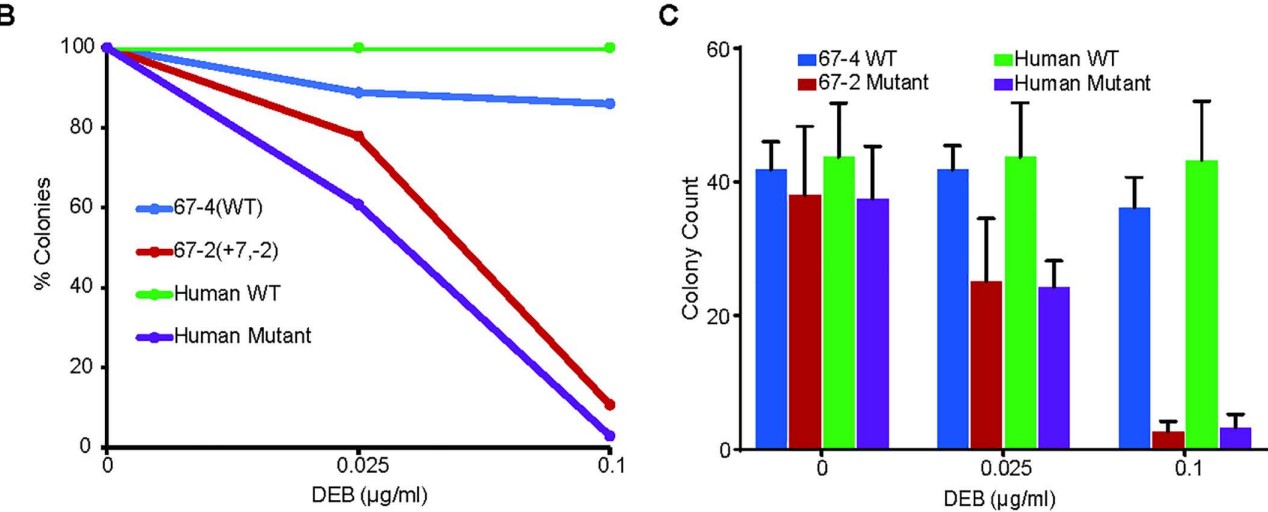

**A**

| ID | Allele 1 | Allele 2 |
|---|---|---|
| 67 - 1 | -2 | WT |
| 67 - 2 | -2 | +7 |
| 67 - 3 | WT | WT |
| 67 - 4 | WT | WT |
| 67 - 5 | -2 | WT |
| 67 - 6 | WT | +7 |
| 67 - 8 | WT | WT |
| 67 - 9 | WT | +7 |

**Fig 5. Evaluation of sensitivity to DNA cross linking in primary cells from a FANCA biallelic exon 4 targeted pig. A)** A colony forming assay was performed on primary fibroblasts from the biallelic piglet 67−2. Cells were plated in duplicate in 6 well plates at the shown concentrations of DEB. A wild-type littermate (67−4) and the transformed human cell line GM639 were used as wild-type controls. The transformed human cell line GM6914 was included as a FANCA-deficient control. Combined data showing total colony counts from 3 independent experiments is shown. Error bars indicate SEM.

while neutropenia is seen in a subset of patients who often go on to develop bone marrow failure. An increase in eosino-phils to about twice the normal number was observed starting at 28 weeks of age; however, routine testing did not reveal any evidence of a parasitic infection.

To assess hematopoietic progenitor function, bone marrow from an edited animal was compared to a control in a hematopoietic colony forming assay (CFC). $FANCA^{-/-}$ pig progenitor cells had a significantly reduced CFC frequency in non-fractionated BM cells of $3.8 \pm 1.4$ per $10^5$ input cells vs a CFC frequency of $14.8 \pm 2.7$ per $10^5$ input cells in controls; ($p = 0.0001$) (**Fig 7B**).

## Strain cryopreservation

The *FANCA* edited pigs were cryopreserved to maintain the line after termination of the experimental animals. A female sibling of 67−2 (67−1, a monoallelic FANCA edited gilt with a WT allele and a 2 bp deletion allele) was bred to founder boar 115−8 (+7 bp insertion) to create a fetal fibroblast cell line that could be used to recreate the line by somatic cell nuclear transfer [28]. This mating resulted in 11 fetuses, and 1 fetus was biallelically edited with the same genotype as Under Strain cryopreservation heading The *FANCA* edited pigs were cryopreserved to ...... with the same genotype as

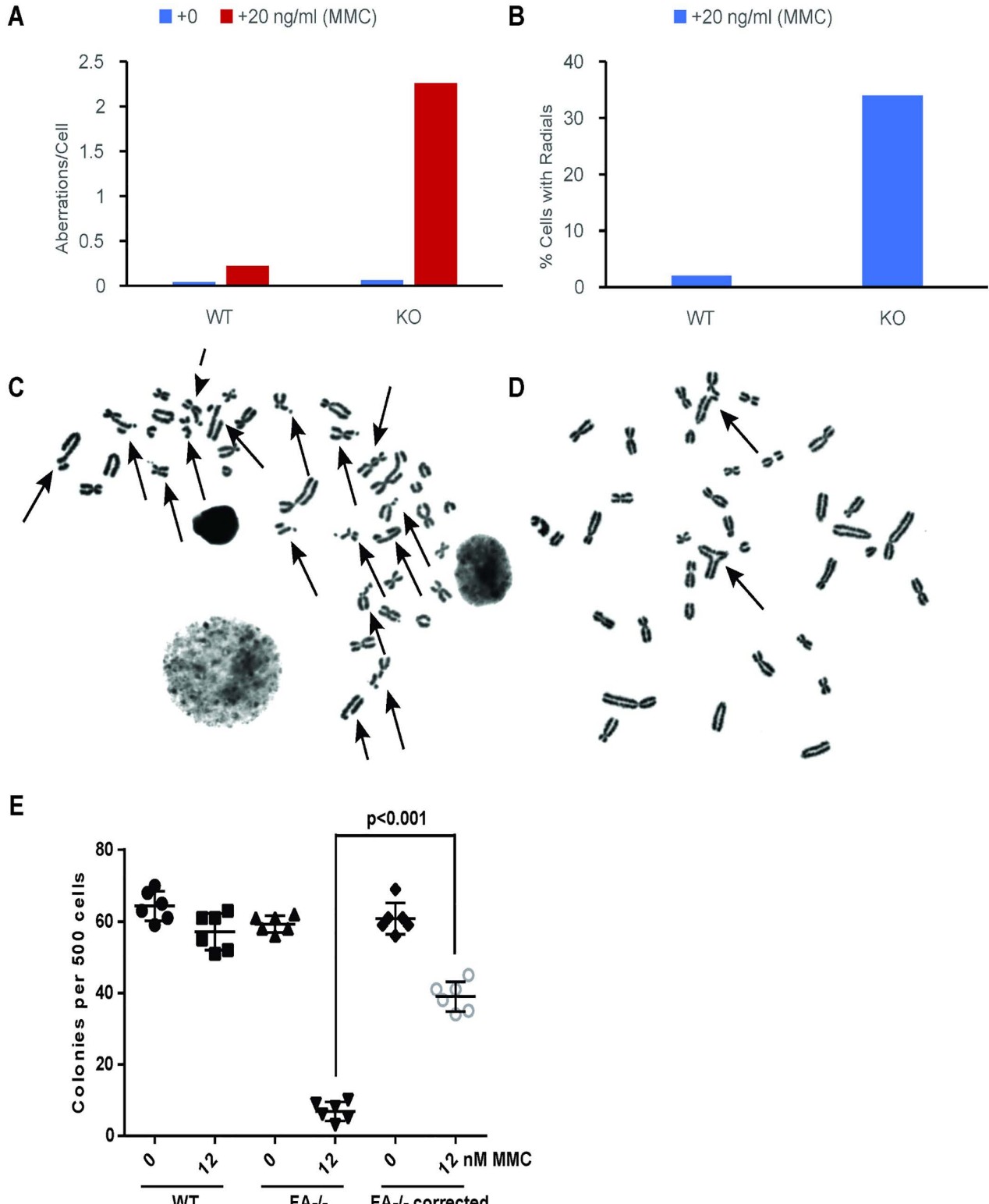

**Fig 6. MMC exposure results in chromosomal aberrations in a biallelic *FANCA* exon 4 targeted pig.** WT and 67−2 [(+7, −2)] peripheral blood mononuclear cells were cultured in the presence of PHA, treated with 20 ng/ml of MMC followed by colcemid arrest and metaphases were harvested for cytogenetic analysis. **A)** Quantification of aberrations/cell in wild type and *FANCA* KO pig lymphocytes exposed to either 0 or 20 ng/ml MMC treatment for 48h.

**B)** Percent of cells with radials in *FANCA* KO and WT pig lymphocytes treated with 0 or 20 ng/ml MMC. **CD)** Examples of chromosomal aberrations and radials, as indicated by arrows, in metaphase spreads from *FANCA* KO pig lymphocytes post MMC treatment. **(E)** The frequency of colony formation by cultured tail fibroblasts derived from a control pig and the 67−2 *FANCA⁻ᐟ⁻* (FA-/-) pig grown in the presence or absence of 12nM MMC. Fibroblasts from 67−2 were also transduced with a lentivirus expressing human FANCA (FA-/- corrected) prior to MMC treatment. Mean and SEM is indicated (p value determined using a student's t-test.

**A**

| Parameter | 8 Wks | 16 Wks | 28 Wks | 38 Wks | 48 Wks | 63 Wks | Normal Range |
|---|---|---|---|---|---|---|---|
| WBC | 12.8 | 17.9 | 17.4 | 16.7 | 14.0 | 13.5 | 11.3-22.8 (x10³/µl) |
| RBC | 6.3 | 6.1 | 6.8 | 7.3 | 6.9 | 6.9 | 5.9-8.8 (x10⁶/µl) |
| HGB | 11.9 | 12.1 | 13.8 | 15.0 | 14.6 | 15.1 | 10.1-15.1 (g/dl) |
| HCT | 37.8 | 36.9 | 40.3 | **46.7*** | 43.7 | 44.6 | 31.1-45.9 (%) |
| MCV | 60.4 | 60.7 | 59.6 | **65.6*** | **63.7*** | **64.7*** | 44.2-60.9 (fl) |
| MCH | 19.1 | 19.9 | **20.4*** | **20.6*** | **21.3*** | **21.9*** | 14.4-20.1 (pg) |
| NEUT | 3.3 | 5.0 | 5.8 | **2.2†** | **2.4†** | **2.6†** | 3.1-9.6 (x10³/µl) |
| LYMPH | 8.6 | **10.9*** | 8.5 | 11.9* | 9.3 | 8.1 | 4.6-10 (x10³/µl) |
| MONO | 0.9 | **1.3*** | **1.2*** | 0.5 | 0.4 | 0.8 | 0.3-1.2 (x10³/µl) |
| EOS | 0.0 | 0.7 | **1.9*** | **1.3*** | **2.0 *** | **2.0*** | 0.0-0.9 (x10³/µl) |
| PLT | 436 | 278 | 265 | 278 | 382 | 302 | 138-468 (x10³/µl) |

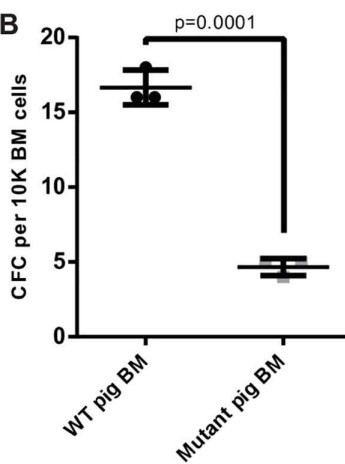

**B** p=0.0001

**Fig 7. FANCA⁻ᐟ⁻ pigs exhibit hematopoietic defects. A)** CBC profiling at multiple time points shows progressive defects in multiple lineages. **B)** Colony forming cell (CFC) frequency in cryopreserved bone marrow is significantly lower in mutant animals than in normal controls. The mean and standard error of the mean from triplicate plates in a representative CFC assay is shown. Student's t-test P value is indicated. EOS, Eosinophils; HCT, hematocrit; HGB, hemoglobin; LYMPH, lymphocytes; MCH, mean corpuscular hemoglobin; MCV, mean corpuscular volume; MONO, monocytes; NEUT, neutrophils; PLT, platelets; RBC, red blood cell; WBC, white blood cell. Bold indicates outside of normal range. * = High value, † = Low value.

672 that was extensively characterized above. *FANCA* 672 that was extensively characterized above. All cell lines were cryopreserved in 10% DMSO and 90% fetal bovine serum (FBS) and are available for rederivation via the National Swine Resource and Research Center (RRID, NSRRC:0075). Additionally, cell lines were created from all founder pigs from litter 115. Semen from founder boar 115−8 (+7 bp) was also cryopreserved for rederivation by breeding if the fetal cell line is not successfully clonable.

## FANCD2 pig generation

In parallel to *FANCA* exon 4 targeting, we also chose to target *FANCD2* because of the more severe phenotype seen in FANCD2 edited animals [29–31] compared to other FA mouse models. Using a similar CRISPR/Cas9 approach we targeted *FANCD2* in exons 31 and 32. Targeted embryos developed to the blastocyst/morula stage at the expected frequency (data not shown). Of 14 piglets born to 2 sows, 8 were wild type, 3 had large monoallelic deletions and 1 showed probable mosaicism with a second allele that was −3. A single animal, 126−3, had a complex genotype (+146 ex31/ + 1 ex32 and −3) and also appeared normal at birth (S2 Table). Primary fibroblasts from 126−3 were tested in a colony forming assay revealing an intermediate level of sensitivity to DEB (Fig 8). However, peripheral blood lymphocytes did not show any radial or aberrant chromosomes (data not shown). Breeding pairs of FANCD2 targeted pigs (S3 Table), produced four F1 litters with a total of 28 piglets (S4 Table). Although it was anticipated that 7 (25%) piglets would have biallelic out-of-frame mutations, none were detected at birth, a finding consistent with embryonic lethality.

To further evaluate the timing of embryonic death, breeding pairs were set up and embryos were harvested at day 35 and day 26 (S3 Table). At the time of harvest, all embryos appeared normal and there was no evidence of embryo

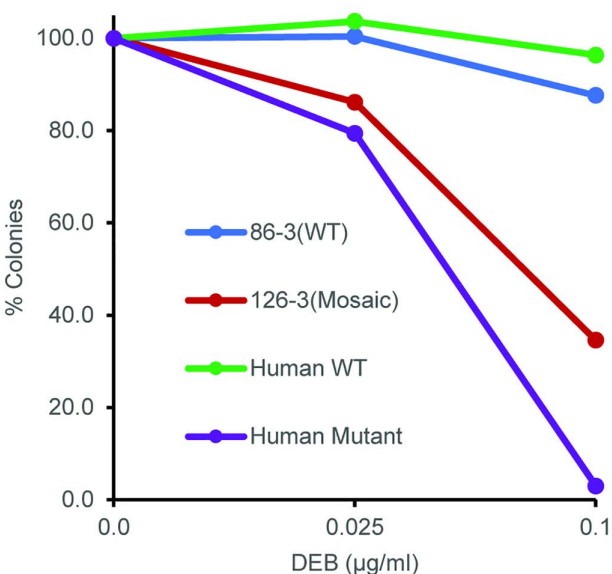

**Fig 8. Sensitivity of mosaic piglet (126−3) to DNA cross-linking agents.** Colony forming activity for fibroblasts from the wild type and mutant human and pig cells following exposure to increasing concentrations of the DNA cross-linking agent DEB is shown.

reabsorption. Genotyping revealed that none of the 19 embryos had biallelic out-of-frame FANCD2 mutations (**S5 Table**). Taken together, these studies revealed no *FANCD2* null mutations in a total of 47 piglets compared to predicted frequency of 25% or ~ 12 animals. These results indicate that an absence of FANCD2 leads to early embryonic lethality after day 7 blastocyst stage of development, but prior to day 26 of gestation.

## Discussion

Our initial studies of the disruption of *FANCA* exon 4 pig fibroblasts in vitro showed the anticipated hypersensitivity to DNA crosslinking agents, similar to the results seen in mouse and human fibroblasts. Based on the results of this proof-of-concept experiment, we used CRISPR CAS 9 to edit exon 4 of the *FANCA* gene in zygotes. Although we obtained piglets with biallelic edits, it was important to create a F1 generation to exclude the potential confounding effects of any off-target effects.

An F1 generation piglet (67−2) showed biallelic out of frame edits (+7,-2) both of which would be predicted to disrupt the function of the FANCA protein. This animal was particularly informative at birth as it had an extra dew claw on both front limbs, a finding not previously reported in domestic swine. Interestingly this skeletal abnormality is similar to the mid-line preaxial polydactyly seen in some human patients with FA [23]. This finding is notable as mouse models of FA do not typically show polydactyly..

Functional characterization of this biallelic FANCA null animal showed that primary skin fibroblasts had hypersensitivity to the crosslinking agent DEB in a colony forming assay. The marked degree of colony forming inhibition was similar to that seen in the human FA fibroblast cell line GM6914. Transduction of mutant fibroblasts with a lentivirus encoding human FANCA significantly reduced the sensitivity of these cells to the alkylating agent MC. These findings confirm the sensitivity to alkylating agents was due to the absence of FANCA activity and not due to an off-target effect.

The gold standard for the clinical diagnosis of FA in humans is the chromosomal breakage assay whereby stimulated T-lymphocytes are treated with MMC and metaphase cells are examined for the presence of chromosomal aberrations and chromosomal radials. Exposure of lymphocytes from our biallelic pig showed greater than a 10-fold increase in chromosomal

aberrations per cell with many cells showing the characteristic chromosome radials. The extent of this chromosomal damage was similar to that seen in human FA and in both mouse and zebrafish models of FA. Importantly, these results indicate the DNA damage phenotype in FANCA targeted pigs is functionally equivalent to the disruption seen in FA patients.

Mouse models of FA typically do not recapitulate the hematologic phenotype seen in FA patients [8]. One of the most common clinical features in patients with FA is the development of anemia and over time the loss of other blood cell lineages producing a picture of bone marrow failure. Many of these patients will go on to develop acute myeloid leukemia. Evaluation of steady state hematopoiesis in our FANCA null pig (67.2) revealed that all blood lineages appeared normal until 28 weeks of age when an increased red cell volume was first noted and this remained elevated through 63 weeks of age. Red cell macrocytosis typically precedes the development of anemia in patients with FA. Similarly, the absolute neutrophil count slowly decreased and this decline in neutrophils is commonly seen in FA patients that go on to develop severe bone marrow failure. Analysis of hematopoietic progenitors in the bone marrow at 63 weeks revealed that the frequency of hematopoietic colony forming cells was reduced by ~75%, another finding likely indicative of developing bone marrow failure. Taken together, these results demonstrate that the pig FANCA model replicates many of the features of early bone marrow failure seen in FA patients.

Although FANCA is by far the most commonly affected gene accounting for about 65% of patients with FA, we were also interested in evaluating FAND2 as it typically shows the most severe phenotype in mouse models [32]. Consequently, we targeted FAND2 in parallel to developing the FANCA model. FANCD2 targeted embryos developed normally to the blastocyst/morula stage however, none of the piglets born to 2 sows showed a biallelic disruption of FANCD2. Similarly, none of the 28 piglets from the F1 generation showed biallelic mutations when the expected result was 7 (25% frequency) suggesting that FANCD2 deficiency resulted in embryonic lethality. To investigate this issue further, embryos were harvested at early gestation and none of these showed the expected FANCD2 biallelic out-of-frame mutations indicating the embryonic lethality occurred in very early gestation. Interestingly, these findings fit with human studies that show patients with FANCD2 are actually hypomorphs that produce low levels of the functional D2 protein [33].

In summary, our results demonstrate the proof of concept that porcine models of FA can replicate many of the clinical features of human FA including skeletal abnormalities, compromised hematopoiesis and abnormal proliferation and differentiation of oral epithelium. Long -term studies of these models are needed to evaluate the possible development of acute leukemia and squamous cell carcinoma, two important complications seen in patients with FA but not typically observed in mouse models. The FANCA (exon 4 + 7,-2) porcine model hold promise for the development of strategies to prevent the long-term complications in FA patients. Mutant fetal fibroblasts have been preserved and could serve to generate cohorts of genetically identical FANCA edited pigs for CRISPR mediated gene therapy. In addition, the controlled ablation of the bone marrow in FANCA edited pigs combined with the transplantation of human hematopoietic stem and progenitor cells could potentially provide a large animal model of human hematopoiesis.

## Supporting information

**S1 Table. FANCA exon 4 targeted FANCA pigs.** Outcome of initial targeting is shown.
(DOCX)

**S2 Table. FANCD2 Exon 31/32 targeted F0 pig generation.** Offspring from IVF treated sow 126 and 127 are shown. *Could not identify mutations through sequencing.
(DOCX)

**S3 Table. FANCD2 exon 31/32 targeted breeding pairs.** * 126−2 has an additional mosaic −3 mutation in exon 32. †126−3 has an additional mosaic −86 mutation in exon 31 and +226 mutation in exon 32. Fetal harvest at day 35 of gestation. Fetal harvest at day 26 of gestation.
(DOCX)

**S4 Table. FANCD2 F1 generation pig litters.** *Litter 46–126−2 x 126−6; Litter 59–126−3 x 126−6; Litter 23–127−2 x 126−7; Litter 86–126−3 x 126−6.*
(DOCX)

**S5 Table. FANCD2 exon 31/32 F1 generation fetal genotypes.** Pig fetuses harvested in early gestation are shown.
(DOCX)

## Author contributions

**Conceptualization:** Kristin M. Whitworth, Markus Grompe.

**Data curation:** Kristin M. Whitworth, Lisa Moreau.

**Formal analysis:** Brandon Hergert, Kristin M. Whitworth, Devorah C. Goldman, Alan D'Andrea, Markus Grompe.

**Funding acquisition:** Randall S. Prather, Markus Grompe, William H. Fleming.

**Investigation:** Brandon Hergert, Kristin M. Whitworth, Devorah C. Goldman, Lisa Moreau, Kelsey McQueen, Kalindi Parmar, Melissa S. Samuel, Craig Dorell, William H. Fleming.

**Methodology:** Devorah C. Goldman, Kevin D. Wells, Randall S. Prather.

**Project administration:** Alan D'Andrea, Randall S. Prather, William H. Fleming.

**Resources:** William H. Fleming.

**Supervision:** Devorah C. Goldman, Kalindi Parmar, Alan D'Andrea, Markus Grompe, William H. Fleming.

**Writing – original draft:** Brandon Hergert.

**Writing – review & editing:** Kristin M. Whitworth, Devorah C. Goldman, Craig Dorell, Markus Grompe, William H. Fleming.

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
