## [Decision Letter · Decision Letter 0]

9 Jul 2025

We look forward to receiving your revised manuscript.

Kind regards,

Younghoon Kee, Ph.D.

Academic Editor

PLOS ONE

Journal Requirements:

2. To comply with PLOS One submissions requirements, in your Methods section, please provide additional information regarding the experiments involving animals and ensure you have included details on (1) methods of sacrifice, (2) methods of anesthesia and/or analgesia, and (3) efforts to alleviate suffering.

“U420D011140 National Heart Lung & Blood Inst. (NIH) RSP

No # Fanconi Anemia Research fund MG and WHF Co-PIs”

7. Please include a copy of Table 1 which you refer to in your text on page 14 in PDF file.

8. We notice that your supplementary tables are included in the manuscript file. Please remove them and upload them with the file type 'Supporting Information'. Please ensure that each Supporting Information file has a legend listed in the manuscript after the references list.

Reviewers' comments:

Reviewer's Responses to Questions

**Comments to the Author**

1. Is the manuscript technically sound, and do the data support the conclusions?

Reviewer #1: Yes

Reviewer #2: Yes

2. Has the statistical analysis been performed appropriately and rigorously?

Reviewer #1: No

Reviewer #2: Yes

3. Have the authors made all data underlying the findings in their manuscript fully available?

Reviewer #1: Yes

Reviewer #2: Yes

4. Is the manuscript presented in an intelligible fashion and written in standard English?

Reviewer #1: Yes

Reviewer #2: Yes

Reviewer #1: Hergert et al describe the characterisation of a porcine model of Fanconi anemia. FANCA and FANCD2 knockouts were attempted with only the former being viable. The authors appropriately contextualise their work by highlighting the limitations of mouse models in reproducing key clinical phenotypes of FA. CRISPR targeting and genotyping appear technically sound, and the authors show DNA repair defects using gold-standard FA diagnostic assays (DEB sensitivity, chromosomal breakage) and rescue experiments with lentiviral FANCA confirm specificity. This latter experiment shows the utility of the animals to future experiments exploring gene therapy for FA.

The detailed characterisation of progressive haematologic changes appears largely based on a single biallelic animal (pig 67-2), which significantly limits the strength of conclusions. The extra digit observed in this animal suggests it has one feature of FA, but it can only be considered as a “case report” rather than a statistically supported finding. Moreover, the point of using a pig model was to overcome the limitations of a short lifespan in seeing FA phenotypes develop in mice. But mice routinely live well beyond 63 weeks, so this represents a missed opportunity to actually test what the authors set out to do – which is see whether phenotypes might develop later in life for an animal with longer lifespan. The in vitro characterisation of cells derived from the FANCA mouse (several animals used here improves the statistical comparisons of the results).

Despite limitations, this study shows significant potential. The technical achievement of creating a large animal FA model is noteworthy, and the preliminary phenotypic characterisation suggests these animals may inded better model human FA.

In Figure 6B, an improper statistical test is performed. Even though it is not outlined what statistical test was used, the compared values are not independent, but multiple resampling of the same animal over time.

Please indicate whether the founder homozygous FANCA animals had any unusual birth defects such as the extra digit.

In the discussion it is noted that “mouse models of FA do not show skeletal defects” however this is incorrect. See 10.1016/j.semcdb.2020.11.010 for a discussion of the various examples that are published.

It is also noted that “Mouse models of FA typically do not recapitulate the hematologic phenotype seen in FA patients”. While this is true in that most FA mice do not get spontaneous bone marrow failure, they do often have >80% loss of HSCs at a similar 1year of age to the pig analysed in this study. Moreover, they can be induced to an FA-like BMF on aldehyde metabolism mutant backgrounds, or when induced with genotoxins. In this sense, the pig seems no better than the mouse! It does not get BMF either. Rather than concentrate on the unsupported claim (also in the introduction) that the pig might be a better model than the mouse for studying phenotypes, it should focus on how a large animal is better suited to studying gene therapy or other life prolonging therapies. The discussion should also more explicitly reference some of the potential clinical trials utilising gene therapy and gene editing, for which this animal would be useful.

The work merits publication with revisions addressing the statistical analysis limitations and more explicit discussion of the study's constraints. The establishment of this model system represents a valuable contribution to the FA research community, despite the preliminary nature of the current characterisation.

Reviewer #2: In this manuscript, Hergert et al. report the first porcine model of Fanconi anemia (FA), marking an advancement in the field. While some FA mouse models, such as the FANCP knockout, have exhibited characteristic phenotypes observed in humans, most knockout mouse models for FA genes have not faithfully recapitulated human FA pathology. This limitation is likely attributable to the relatively short lifespan and species-specific differences in DNA repair mechanisms. As reliable animal models are essential for dissecting FA pathophysiology and evaluating therapeutic strategies, the development of alternative models has been a pressing need.

In this study, the authors successfully generated a porcine FA model that exhibits hallmark phenotypes of the disease, including congenital abnormalities, hypersensitivity to DNA crosslinking agents, and hematologic defects. These phenotypes closely parallel those seen in human FA patients, underscoring the translational value of this model. The manuscript is easy to read and accessible, and the data are reliable and significant to the field of FA research. However, several formatting and figure presentation issues should be addressed to improve clarity and readability.

Major Issues:

- It is very interesting to observe that FANCD2 knock-out mice are lethal. However, it would be interesting to show if the depletion of FANCD2 in normal porcine fibroblasts presents typical cellular defects of Fanconi anemia as human cases.

Minor Issues:

- Figure Legends and Content Alignment: The presentation of Figure 1A is unclear. In the Results section, the authors begin by validating exon 4 targeting in wild-type porcine fibroblasts. However, Figure 1A does not indicate the specific cell type used in this experiment. Clarification in the figure legend and labeling would enhance interpretability.

- Figure Ordering: The order of figures appears disorganized. For example, the appearance of Figure 1A, then Figure 2A, followed by Figure 1B, disrupts logical flow. Figures should be reorganized to follow the sequence of experiments as presented in the text.

- Figure Quality: The resolution of several figures, particularly Figure 5, is suboptimal. High-resolution versions should be provided to allow proper evaluation of the data.

- Figure Consistency: There are inconsistencies in figure sizing and formatting. Some figures are disproportionately large compared to others. A standardized format should be adopted across all figures to maintain a cohesive presentation.

**Do you want your identity to be public for this peer review?** For information about this choice, including consent withdrawal, please see our Privacy Policy

Reviewer #1: **Yes: ** Andrew J Deans

Reviewer #2: No

---

## [Author Response · Author response to Decision Letter 1]

8 Oct 2025

Please state what role the funders took in the study.

Also please see separate uploaded Response to Reviewers file.

---

## [Editor Report · Decision Letter 1]

16 Oct 2025

A porcine model of Fanconi anemia

PONE-D-25-32259R1

Dear Dr. Fleming,

We’re pleased to inform you that your manuscript has been judged scientifically suitable for publication and will be formally accepted for publication once it meets all outstanding technical requirements.

Kind regards,

Younghoon Kee, Ph.D.

Academic Editor

PLOS ONE
---

## [Editor Report · Acceptance letter]

PONE-D-25-32259R1

PLOS ONE

Dear Dr. Fleming,

I'm pleased to inform you that your manuscript has been deemed suitable for publication in PLOS ONE. Congratulations! Your manuscript is now being handed over to our production team.

Kind regards,

on behalf of

Dr. Younghoon Kee

Academic Editor

PLOS ONE